

# Improving the prediction accuracy of river inflow using two data pre-processing techniques coupled with data-driven model

Hafiza Mamona Nazir[1,*], Ijaz Hussain[1,*], Muhammad Faisal[2,3], Elsayed Elsherbini Elashkar[4] and Alaa Mohamd Shoukry[4,5]

[1] Department of Statistics, Quaid-i-Azam University, Islamabad, Pakistan
[2] Faculty of Health Studies, University of Bradford, Bradford, United Kingdom
[3] Bradford Institute for Health Research, Bradford Teaching Hospitals NHS Foundation Trust, Bradford, United Kingdom
[4] Arriyadh Community College, King Saud University, Riyadh, Saudi Arabia
[5] KSA workers University, Egypt, KSA, Egypt
[*] These authors contributed equally to this work.

Corresponding authors
Ijaz Hussain, ijaz@qau.edu.pk
Alaa Mohamd Shoukry, aabdulhamid@ksu.edu.sa

## ABSTRACT

River inflow prediction plays an important role in water resources management and power-generating systems. But the noises and multi-scale nature of river inflow data adds an extra layer of complexity towards accurate predictive model. To overcome this issue, we proposed a hybrid model, Variational Mode Decomposition (VMD), based on a singular spectrum analysis (SSA) denoising technique. First, SSA his applied to denoise the river inflow data. Second, VMD, a signal processing technique, is employed to decompose the denoised river inflow data into multiple intrinsic mode functions (IMFs), each with a relative frequency scale. Third, Empirical Bayes Threshold (EBT) is applied on non-linear IMF to smooth out. Fourth, predicted models of denoised and decomposed IMFs are established by learning the feature values of the Support Vector Machine (SVM). Finally, the ensemble predicted results are formulated by adding the predicted IMFs. The proposed model is demonstrated using daily river inflow data from four river stations of the Indus River Basin (IRB) system, which is the largest water system in Pakistan. To fully illustrate the superiority of our proposed approach, the SSA-VMD-EBT-SVM hybrid model was compared with SSA-VMD-SVM, VMD-SVM, Empirical Mode Decomposition (EMD) based i.e., EMD-SVM, SSA-EMD-SVM, Ensemble EMD (EEMD) based i.e., EEMD-SVM and SSA-EEMD-SVM. We found that our proposed hybrid SSA-EBT-VMD-SVM model outperformed than others based on following performance measures: the Nash-Sutcliffe Efficiency (NSE), Mean Absolute Percentage Error (MAPE) and Root Mean Square Error (RMSE). Therefore, SSA-VMD-EBT-SVM model can be used for water resources management and power-generating systems using non-linear time series data.

## INTRODUCTION

Reservoirs are recognized as one of the most powerful tool in integrated water resources management. They are considered the major solution in water-related problems like urban and industrial water supply, hydro-power generation, irrigation, flood control and conservation of ecology (*El-Shafie et al., 2008*). However, the reservoir system is a challenging problem due to its complexity as reservoirs should neither be too empty to operate nor too filled with water to allow capture of flood water (*Amnatsan, Yoshikawa & Kanae, 2018*). A reservoir's optimized operation depends on the accuracy of river inflow prediction, which is an essential element not only in reservoir operation but also for many hydrological management problems. Accurate prediction results in better decisions such as, flood and drought controls, the supply of drinking water, water resources management and many optimal environmental operations (*El-Shafie et al., 2008*; *Erdal & Karakurt, 2013*; *Zhou et al., 2018*; *Wang, Qiu & Li, 2018*; *Dehghani et al., 2019*). Over the past decades, numerous methods have been developed for accurate river inflow prediction. Literature related to river inflow prediction can be found from these (*Kisi, 2005*; *Easey, Prudhomme & Hannah, 2006*; *Londhe & Charhate, 2010*; *Adnan et al., 2017a*; *Zaini et al., 2018*). These models are broadly classified into three categories: physical-based models, data-driven models, and hybrid models (*Chen et al., 2018*). All these models have been widely used to predict rivers flow and other hydrologic analyses (*Erdal & Karakurt, 2013*; *Hao et al., 2017*; *Chen et al., 2018*; *Darwen, 2019*; *Wang, Qiu & Li, 2018*). Physical-based models extract the inherent behaviors of hydrological variables by conceptualizing their physical process and characteristics. However, physical-based models require a large amount of data and detailed mathematical equations, which may raise the issue of estimating huge parameters and the expensive computational costs (*Chen et al., 2018*). Moreover, due to the unavailability of long hydrological data, especially in developing countries, it is difficult to obtain these parameters, which limits the application of these models. Comparative to physical-based models, Data-Driven (DD) models are further classified into Traditional Statistical (TS) and Artificial Intelligence (AI) models to predict linear and non-linear data, respectively. TS models, also called Box and Jenkins methodology (*Box & Jenkins, 1970*; *Box & Pierce, 1970*), (*Al-Masudi, 2013*) includes the Autoregressive (AR), the Autoregressive Moving Average (ARMA) and the Autoregressive Integrated Moving Average (ARIMA) models are widely applied for predicting river inflow data. *Adnan et al. (2017b)* used the ARIMA model to predict the streamflow. They took monthly streamflow data and concluded that application of ARIMA can be useful in generating precise prediction. However, the disadvantages of TS models are that the river inflow data must be linear which limits the application of these models (*Wang, Qiu & Li, 2018*). To overcome these drawbacks, AI models have been introduced which includes Artificial Neural Network (ANN), Multi-Layer Perceptron (MLP), Generalized Regression Neural Network (GRNN), Adaptive Neuro Fuzzy Inference System (ANFIS) (*Salih et al., 2019*), Multivariate Adaptive Regression (MAR), M5 Model Tree (*Yaseen, Kisi & Demir, 2016*), Support Vector Machine (SVM), Extreme Learning Machine (ELM) (*Yaseen et al., 2016*), fuzzy logic and Radial Basis Neural Network (RBNN) (*Othman & Naseri, 2011*; *Yang et al., 2017*; *Malik & Kumar, 2018*; *Mosavi, Ozturk & Chau,*

*2018*; *Kim et al., 2019*). These AI techniques have been successively applied in hydrology to accurately predict the river inflow/outflow data (*Othman & Naseri, 2011*; *Valipour, Banihabib & Behbahani, 2013*; *Shamim et al., 2016*; *Yang et al., 2017*; *Malik & Kumar, 2018*; *Mosavi, Ozturk & Chau, 2018*). *Yaseen et al. (2016)* evaluated the potential of ELM algorithm to validate its superiority over other AI methods and suggested that ELM model outperform than the other models to predict monthly streamflow. *Yaseen, Kisi & Demir (2016)* investigated the usefulness of three types of regression models i.e., least square-SVM, MAR and M5 model tree to forecast the monthly streamflow. Their study indicated that SVM model generally perform superiors than the other models. Among AI techniques, SVM, as the most widely used method, has been considered an effective tool in solving many non-linear mapping relationships to precisely predict rivers flow (*Garsole & Rajurkar, 2015*; *Adnan et al., 2018*; *Bafitlhile & Li, 2019*), water level (*Behzad, Asghari & Coppola Jr, 2009*) and many other non-linear problems (*Wu & Lin, 2019*). However, all these AI models needs to be carefully optimized as hydrological time series data becomes more and more complex due to rapid climate and other changes. For that purpose, bio-inspired techniques i.e., genetics algorithm, evaluationary programming, differential evolution, etc., are combined with AI methods to optimize their parameters to enhance its precision (*Zheng et al., 2013*). However, there is a drawback for such bio-inspired based AI methods. First, they ignore the multi-scale nature of hydrological data. Second, they do not incorporate with noises, which is inherited part of hydrological data. Developing a single model to predict river inflow data is a challenging task due to its non-stationary, multi-scale and noisiest characteristics (*Yang et al., 2016*; *Yaseen et al., 2017*; *Yu et al., 2017*; *Al-Sudani, Salih & Yaseen, 2019*; *Rezaie-Balf et al., 2019a*; *Rezaie-Balf et al., 2019b*; *Rezaie-Balf, Kisi & Chua, 2019*). Therefore, using the raw river inflow data may not provide useful results, but applying data pre-processing methods may improve the performance of TS or AI techniques known as hybrid models (*Okkan & Serbes, 2013*; *Chitsaz, Azarnivand & Araghinejad, 2016*; *Chen et al., 2018*; *Wu & Lin, 2019*).

In recent years, hybrid models through data pre-processing techniques have received great attention and commonly applied in non-linear, multi-scale and noisiest time series data such as hydro-meteorology, climatology, finance and economic as powerful alternative modeling tools against alone physical-based or DD models (*Chen et al., 2018*; *Zhang et al., 2018*; *Rezaie-Balf et al., 2019b*; *Nazir et al., 2019*; *Wu & Lin, 2019*). Until now, various data pre-processing-based hybrid models have been developed to address these non-linearity issues present in river inflow series. Among all, five main data pre-processing algorithms, i.e., Fourier Analysis, Wavelet Transform (WT) (*Daubechies, 1990*), and SSA (*Golyandina, Nekrutkin & Zhigljavsky, 2001*), are combined with TS and AI methods to form a hybrid model. All data pre processing techniques can be used either to decompose non-linear and multi-scale data into the time-frequency domain or to denoise the time series data. *Rezaie-Balf, Kisi & Chua (2019)* employed EEMD data pre-processing method to enhance the performance of MAR and M5 Model Tree. They demonstrated that EMD-MAR provides more robust results to predict one-day aheaf river flow. Various studies shows that use of WA have gained popularity in handling multi-scale nature of complex hydrological data by combing with NN and other DD methods. *Mouatadid et al. (2019)* explores the use of

WA-Long Short-term memory network (WA-LSMN) for robust irrigation flow forecasting. Their proposed methodology provides appropriate results rather than the satndalone LSMN model. *Nazir et al. (2019)* developed a WA-based hybrid model to predict the river inflow data of four stations and shown that their proposed model was better than the simple ARIMA and ANN models. Later, an EBT approach was be developed to enhance the precision of WA (*Chipman, Kolaczyk & McCulloch, 1997*; *Johnstone & Silverman, 2005*). In the EBT method, a mixture of priors is selected for the distribution of multi-scale components derived from WA. The posterior median is calculated from selected priors to estimate noise free multi-scale components (*To, Moore & Glaser, 2009*). Moreover, use of EMD (*Huang et al., 1998*) and EEMD (*Wu & Huang, 2009*) with DD models also became popular to study the non-stationary complex hydrological data (*Rezaie-Balf, Kisi & Chua, 2019*) However, all data pre-processing techniques have some drawbacks with different aspects to decompose the non-linear, multi-scale and noisiest data. The most widely used WA depends heavily on the selection of wavelet basis function (*Wang, Qiu & Li, 2018*), the application of EMD is limited by its own mathematical mode mixing and sensitivity to denoise property (*Nazir et al., 2019*), and EEMD suffers a strict mathematical theory (*Qian et al., 2019*; *Wu & Lin, 2019*). However, there is a need for developing new hybrid approaches with efficient decomposition methods that predict the non-linear, high irregular and noise-corrupted data with high precision. Several new data pre-processing approaches have been proposed and VMD is commonly used because of its efficient mathematical sound and more precise multi-scale components separation (*Ali, Khan & Rehman, 2018*; *Wu & Lin, 2019*; *Lei, Su & Hu, 2019*). VMD, as a data decomposition method, has been applied in the field of signal processing and wind speed prediction (*Liu, Mi & Li, 2018*; *Lei, Su & Hu, 2019*). *Rezaie-Balf et al. (2019a)* proposed a new hybrid model comprised on Variational Mode Decomposition based ELM (VMD-ELM) to forecast short-term water demand. Their pre-processing method i.e., VMD, provides better results when compared with the simple ANN and ELM models. Later, the performance of VMD is enhanced by coupling with EEMD and Random Forest Algorithms (EEMD-VMD-RFA) (*Rezaie-Balf, Kisi & Chua, 2019*).

In this article, we aimed to develop a novel hybrid model to employ two-phase decomposition based method to efficiently predict the river inflow time series data. Our proposed method comprised on SSA as denoising, VMD as a data decomposition with EBT threshold, and SVM as a prediction method. This work is one of the first attempts known to the authors to use the SSA method as the primary decomposition technique, to enhance the prediction of daily river inflow records with VMD-EBT and SVM

## PROPOSED METHODOLOGY

In this article, a novel hybrid model i.e., SSA-VMD-EBT-SVM is proposed to improve the accuracy of daily river inflow data. The schematic view of proposed methodology is illustrated in Fig. 1. The proposed structure is comprised of denoising, decomposition-threshold, prediction and aggregation steps. In denoising stage, SSA is used to denoise the river inflow data (*Romero et al., 2015*). In the decomposition-threshold stage, VMD

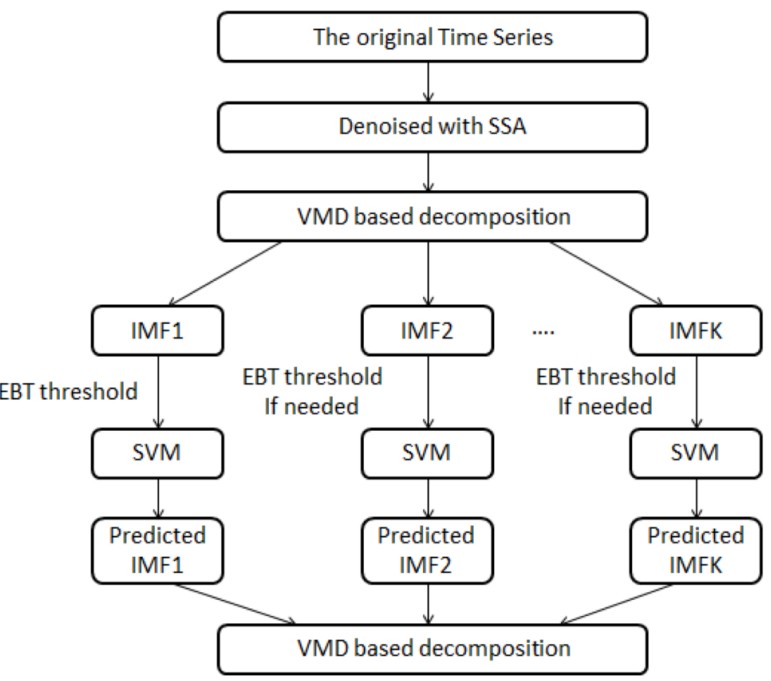

**Figure 1** The schematic view of the proposed model, i.e., the SSA-VMD-EBT-SVM model.

is employed to decompose the denoised daily river inflow series into multiple IMFs (*Rezaie-Balf et al., 2019a*). The high irregular IMF is set as threshold with EBT to remove its sparsity and irregularity (*Nazir et al., 2019*). Further, in the prediction stage, SVM is applied on all IMFs to establish the prediction models and all predicted IMFs are aggregated to get a final prediction (*Yaseen, Kisi & Demir, 2016*). The effectiveness of the proposed hybrid model is evaluated using daily river inflow data from four stations of Indus River Basin (IRB) system, Pakistan (a detail discussion will be in 'Case Study and Experimental Design'). A brief introduction of SSA, VMD, EBT, and SVM is outlined as follows:

## SSA for denoising

For time series analysis, the SSA method is known as a powerful non-parametric method (*Golyandina, Nekrutkin & Zhigljavsky, 2001*). SSA combines the principals of time series analysis, multivariate statistics, dynamical and signal processing (*Suhartono et al., 2018*). The reason for using SSA as it is a model-free technique (*Romero et al., 2015*), which can be applied on any type of data without any assumption. The main function of SSA is to decompose the time series data into a trend, seasonal, oscillations, and aperiodic noises and then reconstruct it after removing aperiodic noises from time series data (*Traore et al., 2017*). Unlike other methods of time series analysis, SSA assumes no statistical assumption about noises while performing analysis and investigating its properties (*Traore et al., 2017*).

### Principles of SSA method

The principle of SSA lies in two stages of decomposition and reconstruction, briefly described as follows:

Consider a time series data $Y_1, Y, \ldots, Y$ of length $N$. The SSA transfer one-dimensional time series data into multi-dimensional $Y_1, Y, \ldots, Y_K$ where $Y_i = (y_1, y, \ldots, y)^T$ and $K = N - L + 1$. These vectors are grouped into a trajectory matrix as:

$$Y = \begin{bmatrix} y_1 & \cdots & y_K \\ \vdots & \ddots & \vdots \\ y_L & \cdots & y_N \end{bmatrix} \tag{1}$$

called Hankel matrix whose all the diagonal elements $i + j = cons$ are equal. The only single parameter in this stage is window length $L$ where $2 < L < N$ (*Traore et al., 2017*). The SSA explores the empirical distribution of pairwise-distances between lagged vectors. The optimality of the SSA method heavily lies on the selection of a window length $L$ as it determines the quality of decomposition (*Traore et al., 2017*). To remove noises from original time series values, eigenvalues are calculated from trajectory matrices which can be written as:

$$Y = E_1, E_2, \ldots, E_d \tag{2}$$

where $d$ is the number of non-zero eigenvalues in decreasing order $(\lambda_1, \lambda_2, \ldots, \lambda_d \geq 0)$ of the $L * L$ matrix of $S = YY$ and $E_i$ is calculated as:

$$E_i = \sqrt{\lambda_i} U_i V_i^T \tag{3}$$

where $i = 1, 2, \ldots, d$, $U_i$ is eigenvectors and $V_i$ is calculated as following:

$$V_i = Y^T \cdot \frac{U_i}{\sqrt{\lambda_i}}. \tag{4}$$

The first few matrices $E_i$ to Y contributed much larger than that of the last few matrices as it is likely that these last matrices represents noises in time series data (*Traore et al., 2017*). The next step is to partition the set of indices i.e., $i = 1, 2, \ldots, d$ into $m$ disjoint subsets i.e., $l_1, l_2, \ldots, l_m$ (*Romero et al., 2015*). Let one of these partitions $I = i_1, \ldots, i_p$, then the trajectory matrix of $I$ is defined as $E_I = E_{i1}, \ldots, E_{ip}$. Once the matrices have been calculated for all partitions, then the original time series trajectory matrice is calculated from these partition matrices as $Y = E_I = E_{i1}, \ldots, E_{ip}$. This step is simplified by approximating matrix $Y$ only with first $r$ matrices $Y = E_1, \ldots, E_r$. The previous step needs a simplification of $r$ parameter appropriately (*Romero et al., 2015*). An approximated time series then recovered from these subsets of matrices by taking the average of diagonals (*Romero et al., 2015*, *Traore et al., 2017*).

## VMD as decomposition
The VMD is a non-recursive signal decomposition estimation method introduced by (*Dragomiretskiy & Zosso, 2014*; *Rezaie-Balf et al., 2019a*; *Rezaie-Balf et al., 2019b*). The VMD adaptively decomposes complicated original non-linear, non-stationary and multi-scale signals into band-limited IMFs i.e., $u_k$ with a specific bandwidth in the spectral domain. To achieve a bandwidth of each IMF, the constrained variational optimization

problem is solved as follows (*Dragomiretskiy & Zosso, 2014*):

$$\min_{\{u_k\}, w_k} \left\{ \sum_k \left\| \partial_t \left[ \left( \delta(t) + \frac{j}{\pi t} \right) * u_k(t) \right] e^{-jwkt} \right\|_2^2 \right\}$$
$$s.t \sum_k u_k = f \qquad (5)$$

where $w_k$ is the center frequency of *kth* IMF, $\delta(t)$ is the Dirac function, $t$ is the time script and $k$ is the number of modes. Moreover, $\left( \delta(t) + \frac{j}{\pi t} \right)$ is the Hilbert transformation function which transform $u_k$ into an analytical signal to form a one-side frequency. The spectrum of each mode can be shifted to a base-mode with the $e^{-jwkt}$ term. The above constrained problem is converted into unconstrained by making use of a quadratic penalty term i.e., $\alpha$ and Lagrangian multipliers $\lambda$, which is easier to address described as follows:

$$L(\{u_k\}, \{\omega_k\}, \lambda) = \alpha \sum_k \left\| \partial_t \left[ \left( \delta(t) + \frac{j}{\pi t} \right) * u_k(t) \right] e^{-jwkt} \right\|_2^2$$
$$+ \left\| f(t) - \sum_{k=1}^{K} u_k(t) \right\|_2^2 + \left\langle \lambda(t), f(t) - \sum_{k=1}^{K} u_k(t) \right\rangle \qquad (6)$$

where $\alpha$ denotes balancing parameter. The Eq. (5) can also be solved by an alternate direction method of multipliers. It is implied that updating $u_k$ $\omega_k$ and $\lambda_k$ in two directions is conducive for realizing the analysis process of VMD, and the solutions of $u_k$ $\omega_k$ and $\lambda_k$ can be calculated as follows:

$$u_k^{n+1}(\omega) = \frac{\hat{f}(\omega) - \sum_{i \neq k} \hat{u}^n(\omega) + \left( \frac{\hat{\lambda}(\omega)}{2} \right)}{1 + 2\alpha(\omega - \omega_k)^2} \qquad (7)$$

$$\omega_k^{n+1} = \frac{\int_0^\infty \omega \left| u_k^{n+1}(\omega) \right|^2 d\omega}{\oint_0^\infty \left| u_k^{n+1}(\omega) \right|^2 d\omega} \qquad (8)$$

and

$$\hat{\lambda}_k^{n+1}(\omega) = \hat{\lambda}_k^n(\omega) + \tau \left( \hat{f}(\omega) - \sum_{i \neq k} \hat{u}^n(\omega) \right) \qquad (9)$$

where $\hat{f}(\omega), u_k^{n+1}(\omega), u_k^n(\omega)$ and $\hat{\lambda}(\omega)$ are the Fourier transforms of $f$ and $n$ denotes the number of iterations. The termination condition of VMD is defined as follows:

$$\frac{\sum_k \left\| u_k^{n+1} - u_k^n \right\|_2^2}{\left\| u_k^n \right\|_2^2} < \epsilon \qquad (10)$$

where $\epsilon$ is the tolerance level of the convergence criterion.

From VMD, the IMF $u_k$ is obtained from the entire decomposition process according to the following steps:

1. Set iteration number $n = 1$, and initialize parameters for VMD including $u_k^1$, $\omega^1$ and $\lambda^1$.
2. Using the Eqs. (8) and (9), calculate $u_k^{n+1}(\omega)$ and $\omega_k^{n+1}$.

3.  After calculating $u_k^{n+1}(\omega)$ and $\omega_k^{n+1}$, update Lagrangian multiplier using the Eq. (9).
4.  If the convergence condition of Eq. (10) is met, the iteration will be stopped, otherwise $n$ moves to $n+1$, and again return to step 2. Finally, the IMF are obtained.

### EBT as a threshold

In EBT method, the posterior distribution is derived with the help of prior distribution to remove sparseness and noises from the coefficients derived from wavelets (*To, Moore & Glaser, 2009*; *Nazir et al., 2019*). In this study, we used this wavelet-based denoising method to remove noises and sparseness of VMD based coefficients. EBT method has level-dependent thresholding approach which deals each IMF according to its own distribution. EBT assumes a mixture of prior distributions for *kth* IMF as follows:

$$p_k(IMF_k|\pi_k,\theta) = (1-\pi_k)\delta_0(IMF_k) + \pi_k\gamma(\theta) \tag{11}$$

where $\pi_k$ is the probability of non-zero coefficients of $IMF_k$, $\delta_0(\theta)$ presents the Dirac delta function of zero part of $IMF_k$ and $\gamma(\theta)$ is a density of non-zero part of $IMF_k$. The prior distribution should be chosen in such a way that it belongs to a family of distributions whose tails decays at polynomial rates. In this regard, Laplace distribution, exponential distribution and quasi-Cauchy distribution have been employed for non-zero coefficients of IMFs which are used to estimate noises (*Nazir et al., 2019*). The probabilities and parameters of a mixture of prior distributions are estimated through maximum likelihood estimation. The reason of using a maximum likelihood approach to estimate unknown is that it determine parameters in such a way that appropriately describe the given data (*Hossain, Kozubowski & Podgórski, 2018*). After estimating parameters, the posterior median $\tilde{\theta}_i(IMF_k,\pi_k)$ is calculated from a mixture of prior distribution as follows:

$$\widetilde{F}_1(\mu|imf) = \int_\mu^\infty f_1(\mu|imf)d\mu \tag{12}$$

which is used as an EBT rule for $\tilde{\mu}$ given data (*Johnstone & Silverman, 2005*). Simple hard rule is further applied to estimate noise-free coefficients of IMFs (*Johnstone & Silverman, 2005*; *Nazir et al., 2019*).

## Support vector machine (SVM) as a prediction method

The SVM is a supervised machine learning method that comprised of statistical learning principles for nonlinear classification, function estimation, and pattern recognition applications (*Vapnik, 1998*). After introducing loss function, SVM can be used as a time series forecasting as well (*Yaseen, Kisi & Demir, 2016*; *Sanghani, Bhatt & Chauhan, 2018*). The concept behind the SVM is that it maps the complex non-linear high dimensional data into a high feature space through a nonlinear mapping. After mapping data into a high feature space, linear regression is performed by SVM in that feature space. Let we have a training set consists of $N$ sample points, $\{x_i, y_i\}_i^N$, where $x_i$ is lagged input vector and $y_i$ is the estimated value of a time series data, then the SVM is formulated as follows:

$$Y = f(x) = w^T\phi(x) + b \tag{13}$$

where $\phi(x)$ is a non-linear transfer function projecting the input data into high dimensional space, $w_i$ are the weight vectors and $b_i$ is a bias. Estimating the sampled values with the

range of allowed precision is considered as the problem of finding the minimum value for $\|w\|$. This can be summarized as convex programming:

$$\min\left(\frac{\|w^2\|}{2}+C\sum_{i}^{N}(\xi+\xi^*)\right) \tag{14}$$

subject to

$$\begin{cases} f(x_i)-y_i \le \varepsilon+\xi^* \\ y_i-f(x_i) \le \epsilon+\xi \\ \xi,\xi^* \ge 0 \end{cases} \tag{15}$$

where $C$ is the user-defined penalty coefficient which represents the dispersion between the weights and objective function. The $\xi$ and $\xi^*$ termed as the slack variables which describes how much data exceeds from tolerance. The Lagrangian function is further applied that uses regression function to replace weight vector and $\phi(x)$ given in Eq. (13) as follows:

$$f(x)=\sum_{i}\left(\alpha_i-\alpha_i^*\right)K(x,x_i)+b \tag{16}$$

where $\alpha_i$ and $\alpha_i^*$ are the Lagrangian multipliers and $K$ is called the kernel function. The possible tested kernels includes linear, polynomial, Gaussian and sigmoid kernels which are defined respectively as follows:

$$K(x,x_i)=x.x_i$$
$$K(x,x_i)=(\gamma(x.x_i)+r)^d$$
$$K(x,x_i)=\exp(-\gamma|x-x_i|^2)$$
$$K(x,x_i)=\tanh(\gamma(x.x_i)+r) \tag{17}$$

where $\gamma$ is the structural parameter, $d$ is a polynomial degree and $r$ represents the residuals of the system. Different values of , $\gamma$ and penalty parameter $C$ is used in this study. The quadratic structure of the Eq. (16) is defined as:

$$W\left(\alpha_i,\alpha_i^*\right)=\sum_{i}y_i\left(\alpha_i-\alpha_i^*\right)-\varepsilon\sum_{i}\left(\alpha_i+\alpha_i^*\right)-\frac{1}{2}\sum_{i}\sum_{j}(\alpha_i-\alpha_i^*)(\alpha_j-\alpha_j^*)K(x_i,x_j) \tag{18}$$

With the following constraints:

$$\sum_{i}^{N}\left(\alpha_i-\alpha_i^*\right)=0$$

$$0 \le \alpha_i \le C, i=1,2,\ldots,N$$
$$0 \le \alpha_i^* \le C, i=1,2,\ldots,N. \tag{19}$$
___________________________________________________________

## Evaluation assessment methods

We assessed and compared the predction performance of our proposed hybrid model SSA-VMD-EBT-SVM with other existing models (EMD-SVM, EEMD-SVM, VMD-SVM, SSA-EMD-SVM, SSA-EEMD-SVM, SSA-VMD-SVM) as a benchmark using following four measures: the Nash-Sutcliffe Efficiency (NSE), Mean Square Error (MSE), Root Mean Square Error (RMSE) (*Ghorbani et al., 2018*) and Mean Absolute Error (MAE) (*Yaseen et al., 2018*) with following equations respectively;

$$NSE = 1 - \left[ \frac{\sum_{t=1}^{N}(y_{ot} - y_{pt})^2}{\sum_{t=1}^{N}(\bar{y}_{ot} - \bar{y}_{ot})^2} \right] \tag{20}$$

$$MSE = \frac{\sum_{t=1}^{N}(y_{ot} - y_{pt})^2}{N} \tag{21}$$

$$RMSE = \sqrt{\frac{\sum_{t=1}^{N}(y_{ot} - y_{pt})^2}{N}} \tag{22}$$

$$MAE = \frac{\sum_{t=1}^{N}|y_{ot} - y_{pt}|}{N} \tag{23}$$

where $y_{ot}$ is the observed values, $\bar{y}_{ot}$ is the mean of observed values and $y_{pt}$ is predicted value of model. Moreover, Taylor diagram is used to prepare a visual comprehension with the help of polar plot for the evaluation of modeling results. The Taylor diagram represents the normalization statndard deviation between simulated and observed values with normalized origin and $R^2$ are represented as directional angles (*Darbandi & Pourhosseini, 2018*). The interpretation of Taylor diagram is that an observed point is shown on graph and the closer the simulated performance measures to the observe point, the better the model performance (*Al-Sudani, Salih & Yaseen, 2019*).

## Benchmark models for the evaluation of the proposed hybrid model

The proposed hybrid model i.e., SSA-VMD-EBT-SVM is compared with six benchmark models described as follows:

a. **Without denoising:** this type of existing models comprised on decomposition and prediction stages only in which VMD and two different data decomposition methods i.e., EMD, and EEMD are chosen which decompose non-linear, non-stationary and multi-scale data into multiple IMFs with the different sound of time-frequency components. For prediction, the extracted IMFs through EMD, EEMD, and VMD are predicted with the same prediction method i.e., SVM as used in our proposed hybrid model. Then, the performance of proposed hybrid model i.e., SSA-VMD-EBT-SVM is compared with existing benchmark models i.e., EMD-SVM (*Yu et al., 2017*) and EEMD-SVM (*Rezaie-Balf et al., 2019b*) and VMD-SVM (*Wu & Lin, 2019*).

b. **With denoising:** these models use denoising-decomposition and prediction stages to predict river inflow data. For denoising, SSA is selected with the same decomposition and prediction stages as described in (a). Then, the performance of the proposed hybrid model i.e., SSA-VMD-EBT-SVM is compared with existing benchmark models i.e., SSA-EMD-SVM, SSA-EEMD-SVM, and SSA-VMD-SVM.

**Table 1  Test statistics and critical values of the ADF test for all four rivers of the IRB system.**

| River inflow | Test statistic | Critical values |
| --- | --- | --- |
| Indus river | −2.8482 | 0.2192 |
| Jhelum river | −2.9841 | 0.1617 |
| Chenab river | −3.2363 | 0.0817 |
| Kabul river | −2.8369 | 0.2240 |

## CASE STUDY AND EXPERIMENTAL DESIGN

The largest water system in Pakistan i.e., IRB is considered for application of proposed architecture as the IRB is Pakistan's largest source of power generation, irrigation, and insensible water resource system. Data from its four major rivers are analyzed i.e., the River Indus, the River Jhelum, the River Chenab, and the River Kabul which contributed significantly in the water system of IRB. The reason of selecting these tributaries is that they are facing frequent river flooding each year due to heavy monsoon rain and melting snow or glacier in Pakistan, glacier-covered 13,680 $km^2$ area which is estimated 13% of the mountainous areas of Upper Indus Basin (UIB). Melted water from these 13% areas adds a significant contribution of water in these rivers. Therefore, it is appropriate to use rivers data of IRB as a representative case study for evaluation of the proposed model.

### Data

The daily river inflow dataset used in this study is comprised on 1st January to 31st March for the period of 2015–2019. To the application of proposed objective, the daily inflow of Indus River at Tarbela with its two principal left and one right bank tributaries: Jhelum River at Mangla, Chenab River at Marala and Kabul River at Nowshera respectively are selected. The daily inflow data is measured in 1,000 ft/s which was acquired from the site of Pakistan Water and Power Development Authority (WAPDA).

## RESULTS

Results of the proposed hybrid model i.e., SSA-VMD-EBT-SVM is defined in stages as follows:

*Denoise-stage results:* first, Augmented Dickey-Fuller (ADF) (*Said & Dickey, 1985*) test is applied on river inflow data of all selected case studies to confirm the non-stationarity. For all case studies, results of the ADF test showed that river inflow data is non-stationary in nature with p-values listed in Table 1. Then the original non-stationary data is processed with SSA to improve the quality of river inflow data by reducing the noises. In processing SSA, window length and number of group i.e., $L$ and $m$ parameter must be determined respectively. Here, different values of $L$ are tested and the optimal value i.e., 90 is selected that gives the lowest error rate of actual and denoised series. The value of $m$ is selected according to the eigenvalues of each river inflow.

The eigenvalues of four selected four rivers of IRB system are shown in Fig. 2 for Indus and Jhelum river inflow, which shows that the values of components of 30, 25, 30 and 20 for Indus, Jhelum, Chenab and Kabul river inflow respectively are clearly larger than those
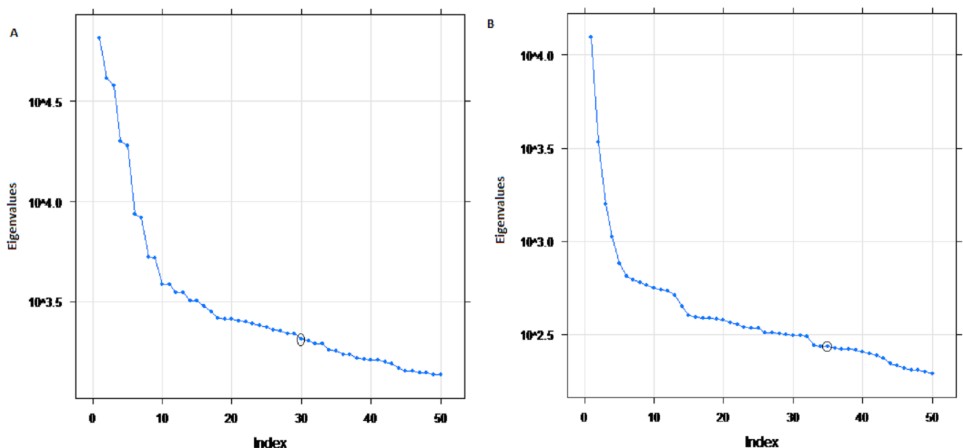

**Figure 2  Eigenvalues of SSA components.** (A) Indus river inflow. (B) Jhelum river inflow.

of the remaining components. The denoised river inflow data is reconstructed by using the selected values of *m*. The processed inflow data for Indus and Jhelum is shown in Fig. 3. The mean and standard deviation of original and denoised river inflow data is listed in Table 2 where it can be observed that mean value remains the same while the standard deviation of denoised river inflow is reduced through processing.

*Decomposition-stage results:* after the original river inflow data is processed with SSA, the denoised data is decomposed into linear and non-linear time-scale oscillations called IMFs by VMD. The number of IMFs i.e., *K* must be selected in advance in order to proceed with any decomposition method. Here, $K = 6$ is selected as the remaining IMFs tend to be similar when $K > 6$. All river inflow data is decomposed into six IMFs. The decomposition results of Indus river inflow is shown in Fig. 4. For comparison, the same river inflow data is also decomposed with EMD and EEMD as shown in Figs. 4 and 5, respectively, for Indus river inflow. It can be seen from Figs. 3–5 that the IMFs extracted through VMD are smoother than the other decomposition methods i.e., EMD and EEMD. However, due to the high oscillations of sixth IMF, extracted through VMD, EBT is applied to denoise IMF. The EBT effectively separate the clear and noisy coefficients of noise dominant IMF by a mixture of prior distributions as defined in Eq. (12) and preserve valid information as much as possible. First, to get normal distribution, the scaled transformation is applied so that each IMF follows $N(\delta_i, 1)$. According to the nature of the sixth IMF as depicted in Fig. 4, the first two IMFs of Fig. 5 and the first three IMFs of Fig. 6 it is known that most of the coefficients in all noisiest IMFs are zero and few are non-zero out of which fewer coefficients are either very low or very high in magnitude. By inspecting both zero and non-zero coefficients of IMFs, a mixture of an atom of probability at zero and different distributions are considered for non-zero part coefficients of IMF (*Johnstone & Silverman, 2005*). Laplace distribution is chosen as a prior distribution out of Exponential and Cauchy distribution for $\delta_i$. Finally, the valid information of IMFs are preserved with posterior median threshold estimator which is calculated through Eq. (13).
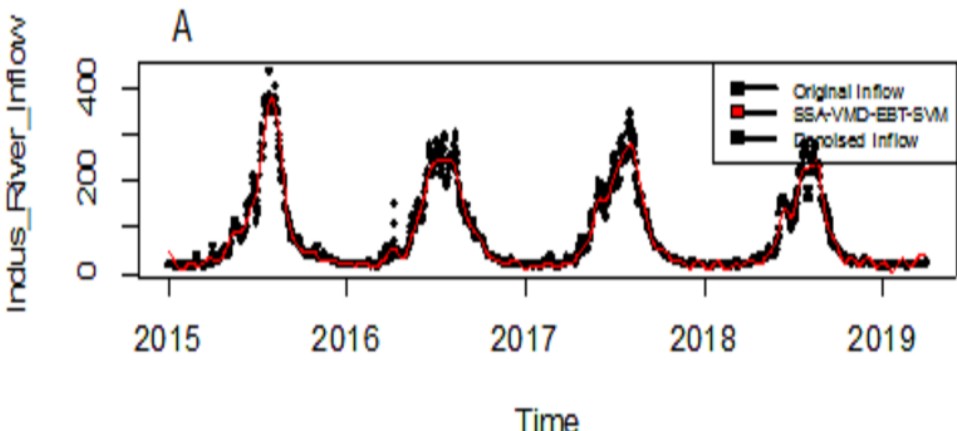

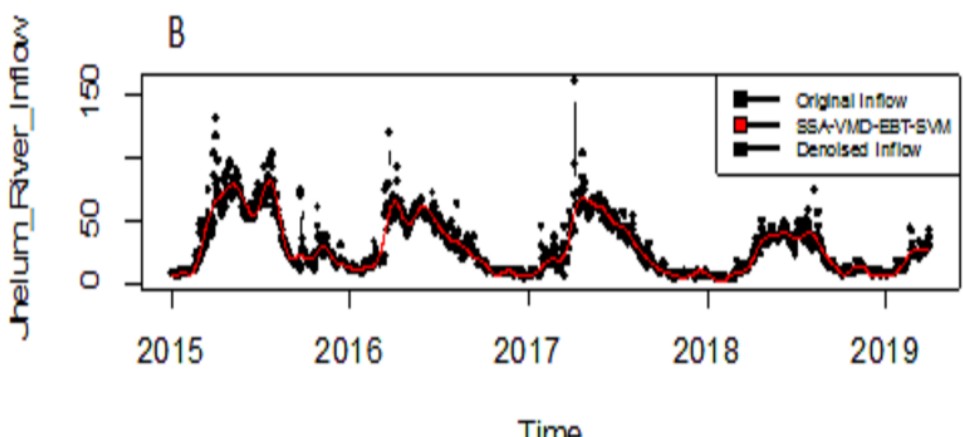

**Figure 3** **Processed inflow data through SSA.** (A) Indus river inflow. (B) Jhelum river inflow.

**Table 2** **Summary statistics of original and denoised rivers inflow data.** SD* shows standard deviation value.

| River inflow | Original inflow | | Denoised inflow | |
|---|---|---|---|---|
| | Mean | SD* | Mean | SD* |
| Indus river | 79.0805 | 86.1221 | 79.9920 | 84.0473 |
| Jhelum river | 28.0692 | 22.5040 | 28.1654 | 21.2862 |
| Chenab river | 27.3528 | 24.6649 | 27.1066 | 23.4877 |
| Kabul river | 31.7901 | 29.4850 | 31.4863 | 27.6831 |

*Prediction results:* finally, the denoised and decomposed IMFs of the proposed model i.e., SSA-VMD-EBT for all selected rivers are predicted through SVM. For that purpose, the daily river inflow data from 1st Jan-2015 to 31 st Dec-2018 i.e., 1,461 observations are used for training and 1st Jan-2019 to 31 st Mar-2019 i.e., 90 observations are used for testing

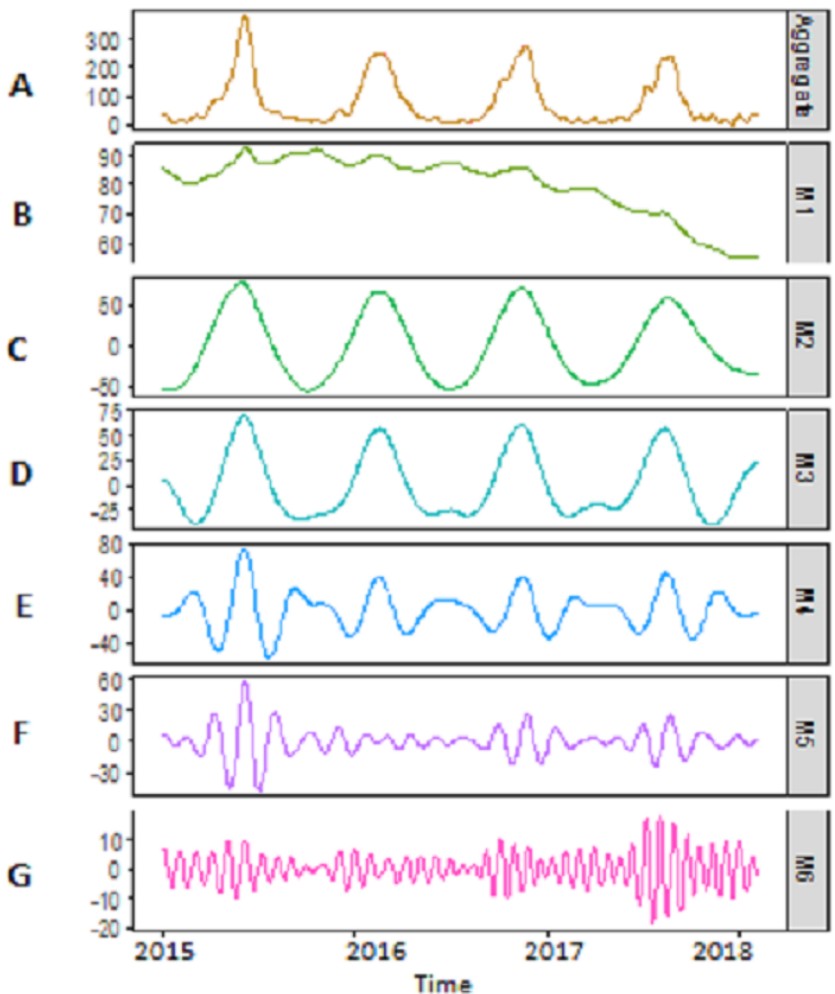

**Figure 4 SSA-VMD decomposition of Indus river inflow.** The series is decomposed into five IMF's (B, C, D, E and F) and one residue (G). The aggregate of five IMF's and residue is represented in A.

purposes. The SVM is trained by setting different values of $\gamma$ and penalty parameter $C$. The values of $\gamma = 10$ and $C = 10,000$ are selected. The parameters of SVM are determined using trial and error basis on which error of training and testing is minimized. according to minimum prediction error for all river inflow. After estimation of each IMF, the accuracy of proposed and benchmark models are measured with NSE, MSE, RMSE, MAE, and MAPE. The performance of proposed hybrid model i.e., SSA-VMD-EBT-SVM is compared with 'without denoising model' i.e., EMD-SVM and EEMD-SVM and VMD-SVM and 'with denoising models' i.e., SSA-EMD-SVM, SSA-EEMD-SVM, and SSA-VMD-SVM. The training results of proposed model i.e., SSA-VMD-EBT-SVM with comparison to all benchmark models for Indus, Jhelum and Chenab river inflow data are listed in Table 3 and results of Kabul river inflow data is presented in Table 4. Moreover, Taylor diagram, as shown in Fig. 7, is used to illustrate the efficiency of proposed model. The graph shows that the proposed model performed very well over other exisiting models. From Tables 3

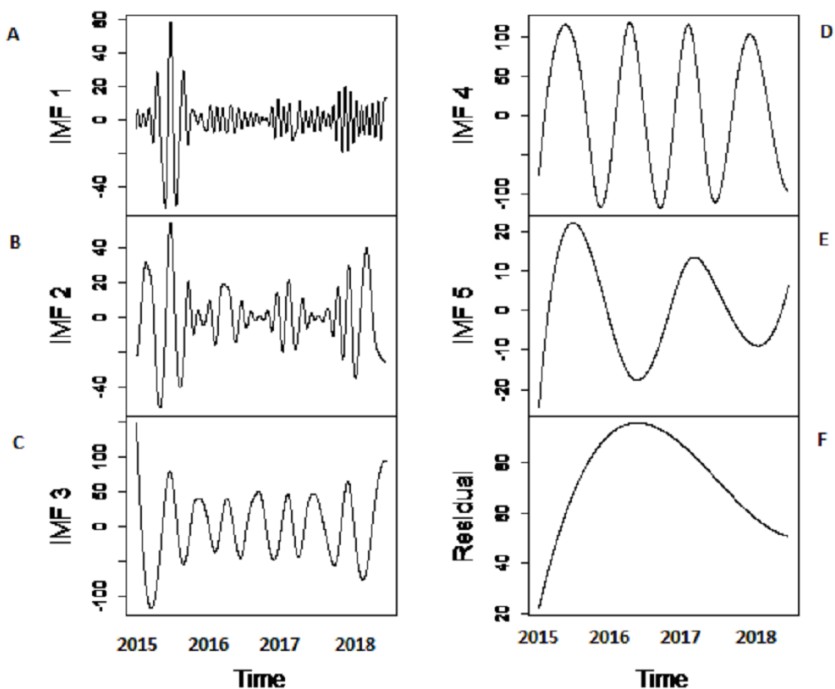

**Figure 5** **The SSA-EMD based decomposition of Indus river inflow.** The series is decomposed into five IMFs (A-E) and one residue (F).

and 4 and Fig. 7, it is concluded that the proposed model results fully demonstrate the effectiveness for all four cases studies with minimum NSE, MSE, RMSE, MAE, and MAPE compared to all with and without denoising models. The worst prediction models are those which are handled through without denoising i.e., EMD-SVM and EEMD-SVM. The validation graphs of proposed model i.e., SSA-VMD-EBT-SVM with comparison to with denoising models i.e., SSA-EMD-SVM, SSA-EEMD-SVM and SSA-VMD-SVM and without denoising models i.e., EMD-SVM and EEMD-SVM and VMD-SVM for Indus and Jhelum river inflow are shown in Fig. 8. From graph, it can be observed that our proposed model i.e., SSA-VMD-EBT-SVM performed well not only in prediction stage but also in validation. However, the exisiting models perform only good in prediction stage as shown in Fig. 8.

## Discussion

In this article, we proposed a novel hybrid model to efficiently predict the river inflow time series data. Our proposed method comprised of SSA as denoising, VMD as a data decomposition with EBT threshold, and SVM as a prediction method.

In order to understand the applicability of our proposed model i.e., SSA-VMD-EBT-SVM, two different model strategies are adapted (see Tables 3 and 4). First, without denoising model strategy is implemented on which the same decomposition method i.e., VMD and two different decomposition methods i.e., EMD and EEMD are used. Moreover, for prediction purpose, SVM which is also adapted in proposed methodology is used

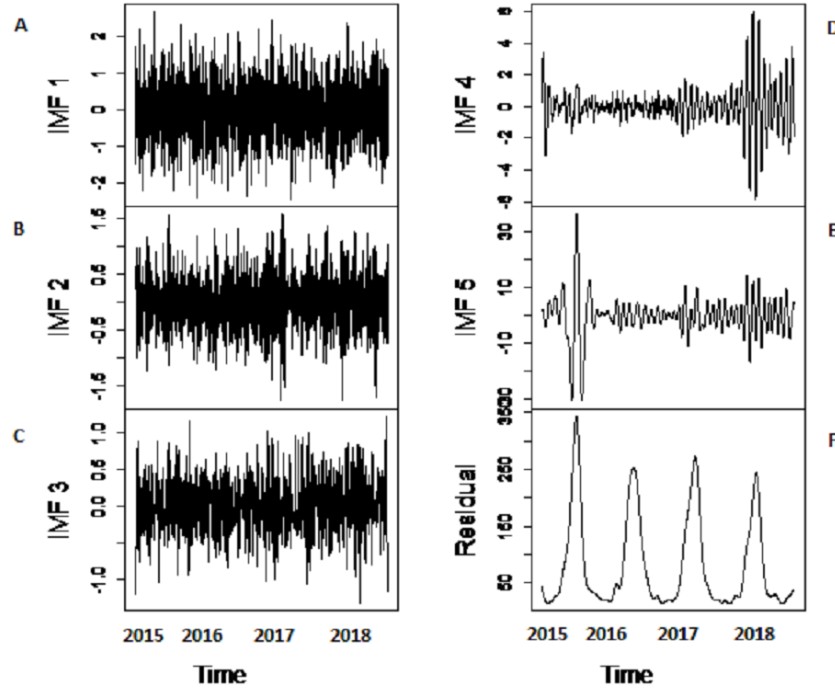

**Figure 6** **The SSA-EEMD based decomposition of Indus river inflow.** The series is decomposed into five IMFs (A-E) and one residue (F).

here to compare the performance of proposed denoised-decomposed strategy with only 'without denoising models i.e., VMD-SVM (*Wu & Lin, 2019*), EMD-SVM (*Yu et al., 2017*) and EEMD-SVM (*Rezaie-Balf et al., 2019b*). From Tables 3 and 4, it can be seen that the overall performance of without denoising models are poor for all river inflow data with high NSE, MSE, RMSE, MAE and MAPE values. Specifically, EMD-SVM performed worst among all without denoising models due to the fact that EMD suffers from the mode mixing problem and fail to produce noiseless IMFs (*Di, Yang & Wang, 2014*). The proposed model performed well with the lowest NSE, MSE, RMSE, MAE and MAPE values as compared to without denoising models.

The second strategy used the concept of denoising and decomposition, here the same method of denoising which is used in proposed methodology is employed with different decomposition methods i.e., SSA-EMD-SVM, SSA-EEMD-SVM and with same decomposition method but without thresholding is adapted i.e., SSA-VMD-SVM. From Tables 3 and 4, it is observed that the proposed model i.e., SSA-VMD-EBT-SVM performed well for Indus, Jhelum, and Chenab, but for Kabul river inflow the results of SSA-VMD-EBT-SVM and SSA-VMD-SVM are same as thresholding the IMF did not enhance the prediction performance of SVM. Moreover, SSA-EEMD-SVM also performs well among existing with and without denoising methods for all case studies. Overall, the proposed SSA-VMD-EBT-SVM model showed a much better agreement between predicted and observed river inflow data which demonstrates the suitability of SSA, VMD, and EBT in pre-processing inputs/output data over other decomposition methods i.e., EMD and

**Table 3  Evaluation index of the training prediction error of the proposed model (SSA-VMD-EBT-MM) with all selected models for Indus, Jhelum and Chenab river inflow.** NSE and MAPE are unit free measures whenever, MSE, RMSE and MAE are measure in 1,000 ft/s. The values of the proposed model SSA-VMD-EBT-SVM are represented as bold.

| Station | Models | NSE | MSE | RMSE | MAE | MAPE |
|---|---|---|---|---|---|---|
| Tarbela | EMD-SVM | 0.9946 | 41.2868 | 6.4255 | 5.8657 | 16.9818 |
| | EEMD-SVM | 0.9962 | 39.3084 | 5.4137 | 4.5540 | 11.3814 |
| | SSA-EMD-SVM | 0.9953 | 34.0684 | 5.8368 | 4.9262 | 12.9763 |
| | SSA-EEMD-SVM | 0.9962 | 28.0560 | 5.9551 | 4.9610 | 12.2437 |
| | VMD-SVM | 0.9911 | 68.2308 | 8.2602 | 6.8929 | 19.5466 |
| | SSA-VMD-SVM | 0.9954 | 33.5565 | 5.7928 | 5.2533 | 15.60666 |
| | **SSA-VMD-EBT-SVM** | **0.9965** | **32.7355** | **5.7215** | **4.7800** | **10.9320** |
| Mangla | EMD-SVM | 0.9396 | 31.6725 | 5.6278 | 3.2197 | 16.5250 |
| | EEMD-SVM | 0.9753 | 12.9495 | 3.5985 | 2.2437 | 12.1502 |
| | SSA-EMD-SVM | 0.9912 | 4.1259 | 2.0312 | 1.8052 | 17.0294 |
| | SSA-EEMD-SVM | 0.9769 | 10.8790 | 3.2984 | 2.4393 | 17.4766 |
| | VMD-SVM | 0.9712 | 15.1061 | 3.8866 | 2.4723 | 14.8790 |
| | SSA-VMD-SVM | 0.9731 | 12.6437 | 3.5558 | 2.7425 | 21.2450 |
| | **SSA-VMD-EBT-SVM** | **0.9936** | **3.0168** | **1.7369** | **1.7979** | **7.9978** |
| Marala | EMD-SVM | 0.9760 | 14.9508 | 3.8666 | 2.1770 | 10.9986 |
| | EEMD-SVM | 0.9893 | 6.6943 | 2.5873 | 2.7507 | 10.8725 |
| | SSA-EMD-SVM | 0.9748 | 14.1228 | 3.7581 | 2.8346 | 48.3824 |
| | SSA-EEMD-SVM | 0.9796 | 11.4645 | 3.3859 | 2.5789 | 28.3756 |
| | VMD-SVM | 0.6525 | 216.738 | 14.72202 | 9.9701 | 47.295 |
| | SSA-VMD-SVM | 0.9780 | 12.3627 | 3.5161 | 2.6684 | 38.8350 |
| | **SSA-VMD-EBT-SVM** | **0.9953** | **6.3790** | **2.5257** | **2.0556** | **10.8084** |

**Table 4  Evaluation index of the training prediction error of the proposed model (SSA-VMD-EBT-MM) with all selected models for Kabul river inflow.** NSE and MAPE are unit free measures whenever, MSE, RMSE and MAE are measure in 1,000 ft/s. The values of the proposed model SSA-VMD-EBT-SVM are represented as bold.

| Station | Models | NSE | MSE | RMSE | MAE | MAPE |
|---|---|---|---|---|---|---|
| Nowshera | EMD-SVM | 0.9156 | 75.6570 | 8.6981 | 4.5491 | 19.2872 |
| | EEMD-SVM | 0.9639 | 32.3744 | 5.6898 | 3.0101 | 12.2574 |
| | SSA-EMD-SVM | 0.9820 | 14.1973 | 3.7679 | 2.9610 | 38.3520 |
| | SSA-EEMD-SVM | 0.9880 | 9.4604 | 3.0757 | 2.4825 | 32.0053 |
| | VMD-SVM | 0.9585 | 37.2792 | 6.1057 | 3.4760 | 17.8121 |
| | SSA-VMD-SVM | 0.9946 | 4.2555 | 2.0629 | 1.8699 | 21.0699 |
| | **SSA-VMD-EBT-SVM** | **0.9945** | **4.2555** | **2.0628** | **1.8699** | **21.0699** |

EEMD. Thus, it is concluded that the appropriate way of denoising, decomposition and thresholding can effectively enhance the performance of non-linear, non-stationary and multi-scale time series data.

By applying proposed simulation models in IRB, it is expected that this will provide new tools for improving inflow prediction over what is possible with the current generation of

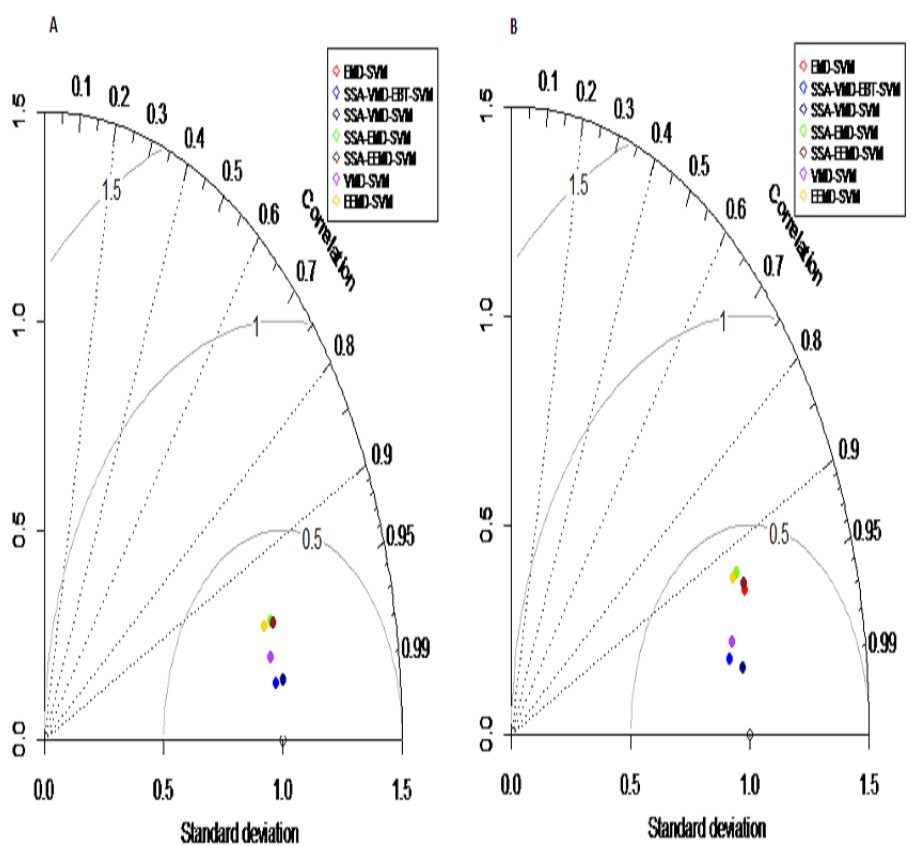

**Figure 7  Taylor diagram: performance measures for the Indus (A) and Jhelum (B) river inflow.**

statistical models as well as help with other land and water management questions. It may also be helpful in setting policies regarding what appropriate methods should be chosen for 'denoising-decomposition and prediction' and in assessing the effects of climate warming. These modeling efforts are therefore significant both for the scientific issues involved as well as for the practical relevance of the results.

## CONCLUSION

The reliable and accurate prediction of river inflow is essential in order to manage water resources. In this article, a hybrid prediction model i.e., SSA-VMD-EBT-SVM is proposed and applied for the prediction of daily river inflow data of four rivers of IRB. The original river inflow data is denoised with SSA and decomposed into several linear and non-linear IMFs by using VMD, then EBT is applied on non-linear IMF to remove noises and sparsities. Finally, each IMF is predicted with SVM and the predicted results of IMF component are aggregated as the final prediction results. To compare the performance of the proposed model, the benchmark model with two different decomposition methods i.e., EMD and EEMD methods combined with SSA-based denoising and without denoising is selected. The five performance indicators NSE, MSE, MAE, RMSE, and MAPE are employed

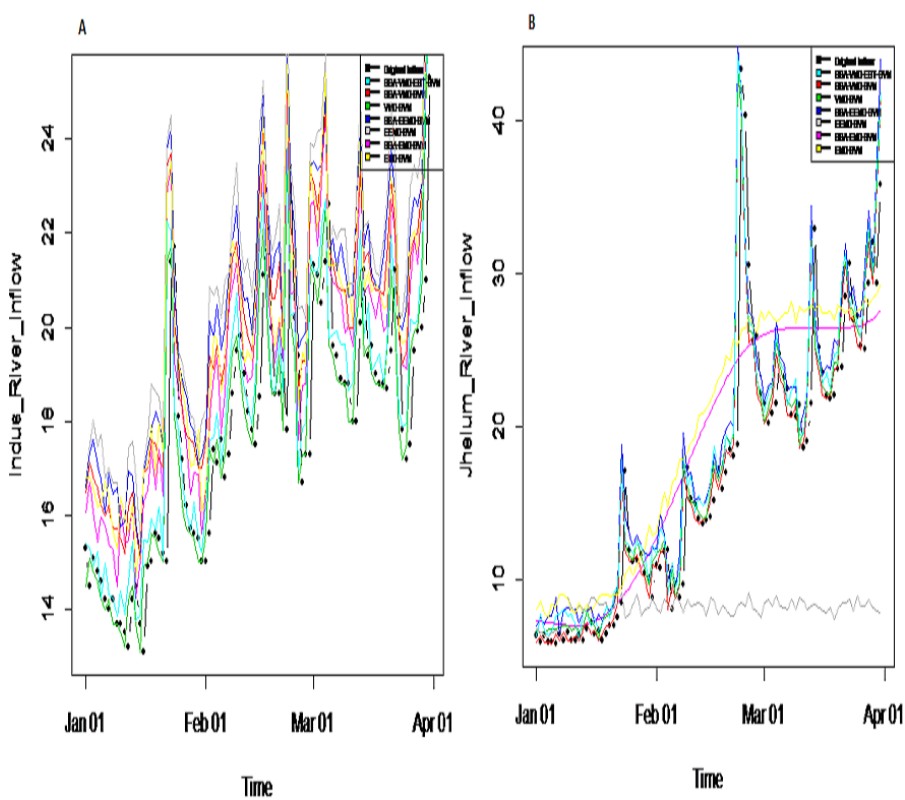

**Figure 8** **Validation graph of proposed SSA-VMD-EBT-MM and all other benchmark models.** (A) Indus river inflow. (B) Jhelum river inflow.

to measure the prediction accuracy of proposed SSA-VMD-EBT-SVM, and all other benchmark models. Based on the results, it is observed that the proposed hybrid model i.e., SSA-VMD-EBT-SVM shown the efficient results with minimum errors. In other words, compared with other models, the proposed hybrid model improves prediction accuracy and reduces errors. The results from this research will not only beneficial for sustainable water resource management but also for other non-linear time series data.

### Funding

The Deanship of Scientific Research at King Saud University funded this work through research group no RG–1439–015. The funders had no role in study design, data collection and analysis, decision to publish, or preparation of the manuscript.

### Grant Disclosures

The following grant information was disclosed by the authors:
Deanship of Scientific Research at King Saud University: RG–1439–015.

## Competing Interests

The authors declare there are no competing interests.

## Author Contributions

- Hafiza Mamona Nazir conceived and designed the experiments, performed the experiments, analyzed the data, prepared figures and/or tables, authored or reviewed drafts of the paper.
- Ijaz Hussain performed the experiments, analyzed the data, contributed reagents/-materials/analysis tools, authored or reviewed drafts of the paper, approved the final draft.
- Muhammad Faisal conceived and designed the experiments, performed the experiments, analyzed the data, approved the final draft, proof readings.
- Elsayed Elsherbini Elashkar conceived and designed the experiments, approved the final draft, final reading.
- Alaa Mohamd Shoukry analyzed the data, prepared figures and/or tables, approved the final draft.

## Data Availability

Raw data is available as a Supplementary Files.

## Supplemental Information

Supplemental information for this article can be found online at http://dx.doi.org/10.7717/peerj.8043#supplemental-information.

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
