# Peer review of "Improving the prediction accuracy of river inflow using two data pre-processing techniques coupled with data-driven model"

_PeerJ, doi:10.7717/peerj.8043_

## Round 0.1 · original submission · Major Revisions

Both reviewers consider that the new hybrid technique for river inflow prediction to be relevant, but both have found considerable issues with the manuscript as presented. I hope that you will be able to address these in a revised version of your manuscript.

Major limitations are highlighted, partly resulting from the way the paper is presented and organised, and particularly resulting from a lack of sufficient detail and justification regarding the methodology and verification. Reviewer 2 additionally finds numerous technical errors, does not consider that the results are treated fairly and, as a result, recommended rejection. In reworking the paper, please ensure that you address these serious concerns.

Several detailed suggestions for improvement have been provided, including references to other research which will help you to improve the paper. Note that, in particular, it is essential that you ensure appropriate and correct application of SSA, and provide sufficient and clear details on how this and other methods were applied to the dataset, and how results were verified. When submitting a new version, please ensure that you provide a response document which provides details of changes you have made to address the issues highlighted.

Reviewer 1 ·

Basic reporting

The paper develops an integrative AI technique to predict inflow pattern. Thus, it is part of the large body of literature related to hydrological and environmental researches. It is commonly known on the field. The most relevant aspect of the paper is the application of a new technique of 'hybrid' model. It is novel enough and seems technically sound and should be considered for publication however, the paper in its present form is still far of being mature enough to be published. The paper needs to improve on organization, elaboration, documentation of procedures and discussion as well as in the English language

Experimental design

I can not comprehend the actual proposed modeling procedure. You need to report a clear flow chart presenting the adopted learning process. Cite appropriate references for the methodology phase. Revise all the notation and formulations. You need to report carefully the the historical data construction for the prediction process. Clearly state the data division.

Validity of the findings

First of all, you need to state the performance formulations. The mathematical expressions. Readers need to be familiar with those metrics. Cite the following references for those indicators:
https://www.sciencedirect.com/science/article/pii/S0022169418307819
https://link.springer.com/article/10.1007/s11269-018-2038-x

Based on the reported statistical metrics. Merely a marginal enhancement are obtained. hence a validation with similar researches picked up from the literature is recommended.

Scatter plot evaluation Taylor diagram are needed to be generated for the attained results.

The discussion phase is most likely a statistical display. I did not see a hydrology interpretation. You need to focus on this. Bear in mind, this is highly essential for your research record.

Additional comments

i. The significant of inflow modeling
ii. The main engineering problem associated with the inlfow problem.
iii. What are the basic methodologies introduced and what are their drawbacks?
iv. What kind of newly developed machine learning models as an intelligence models for inlfow forecasting.?
v. What are the main limitation of these AI models emphasis the hybridization with bio-inspired optimization algorithms?
vi. I do expect the authors report a comprehensive literature review on this. River flow forecasting has been conducted using over 200 scientific researches.
Literature review is very much poorly presented. Hence you need to recall some valid researches and try to discard the conferences papers. Here I suggest you to consider those researches for your literature.
https://www.sciencedirect.com/science/article/pii/S0022169419302471
https://link.springer.com/article/10.1007/s10489-018-1232-0
https://www.sciencedirect.com/science/article/pii/S0378377418311831
https://link.springer.com/article/10.1007/s11269-018-1909-5
https://link.springer.com/article/10.1007/s11269-016-1408-5
https://www.sciencedirect.com/science/article/pii/S0022169418309545
https://www.sciencedirect.com/science/article/pii/S0022169417306029
https://www.sciencedirect.com/science/article/pii/S0022169416305893
https://link.springer.com/article/10.1007/s12665-018-73768
https://www.sciencedirect.com/science/article/pii/S0022169419302513

Reviewer 2 ·

Basic reporting

The most relevant aspect of the paper is the application of a new hybrid technique for river inflow prediction. This research is not so technically sound in methods and analysis. The results are not fairly presented. There are several factors which detract the quality of the manuscript. I cannot recommend publication of this work in its present form. The authors need to address the queries and incorporate the suggestions given below.

Experimental design

1. The authors claim to develop several ensemble or hybrid models for the task of river inflow prediction on daily time scale. They adopt several signal processing methods namely, Variational Mode Decomposition (VMD) based on a Singular Spectrum Analysis (SSA); Empirical Bayes Threshold (EBT); Empirical Mode Decomposition (EMD); and Ensemble EMD for pre-processing of river inflow data. However, it is to be noted that, the manuscript lacks a detailed methodology on how all these techniques were implemented and used for river inflow prediction using SVM.
2. The paper suffers from many technical errors in using the SSA. I have a serious concern on the choice of SSA. It seems like, the authors have decomposed the complete time series at a stretch. It is important to note that performing VMA based SSA decomposition on time series must be independent for each partition periods (training, testing phases) in order to avoid incorporation of information from future data (that is to be used in the testing period) in the training phase.
3. I strongly recommend the following paper for consideration of the authors which gives details on the incorrect usage of SSA. Du, K., Zhao, Y., & Lei, J. (2017). The incorrect usage of singular spectral analysis and discrete wavelet transform in hybrid models to predict hydrological time series. Journal of Hydrology, 552, 44-51.
4. How detailed is description of methods is a key aspect on writing scientific papers. First, note that in the title and in the introduction there are references to novel hybrid modelling and in the section 2 (Lines 123 – 245) there are details related to the mathematical formulations of different methods. In the framework of any novel research that has been carried out, a detailed explanation about what a hybrid modelling is? should be provided, including its origin and the logic to use it. In respect of the link between methods and results the way hybrid modelling is selected is a little bit dark. The paper seems weak in terms of hydrological discussion on the results. The paper needs to improve on organization, elaboration, documentation of procedures (EMD) and discussion.

Validity of the findings

1. If I am not wrong nowhere in the text before is told about this parametric characterization of SVM models. Information cannot be placed so out of context. The authors need to shed light on how the SVM parameters (Cost, Epsilon and Kernel parameter) were optimized?
2. I recommend the authors to use advanced model evaluation metrics such as Gross Performance indicator, Taylor diagrams etc. to present the model results.
3. Clarity of Figure 1 needs to be improved.
4. Tables 3 & 4, mention the units of MAE, RMSE and MSE metrics.

Additional comments

The contribution of manuscript is too weak which unfortunately convinced me to reject the manuscript.

---

## Round 0.2 · accepted · Accept

Thank you for your changes to the manuscript, which have help to clarify your research, and further justify the use of selected methods. While some concerns have been raised regarding the validity of the statistical approach, I consider that it is sufficiently justified in the paper, worthy of the wider scrutiny it can receive through publication, and any responses can be submitted as a reply or through other research papers.

Please proof your manuscript carefully for any typos - e.g. on page 4: "one-day aheaf river flow"

Reviewer 1 ·

Basic reporting

The authors revised the manuscript accordingly.

Experimental design

The methodology is clearly reported.

Validity of the findings

The finding of the research support the main research set-up.

Additional comments

The authors revised the manuscript accordingly.